# The Qualitative and Quantitative Compositions of Phenolic Compounds in Fruits of Lithuanian Heirloom Apple Cultivars

**DOI:** 10.3390/molecules25225263

**Published:** 2020-11-11

**Authors:** Aurita Butkevičiūtė, Mindaugas Liaudanskas, Darius Kviklys, Dalia Gelvonauskienė, Valdimaras Janulis

**Affiliations:** 1Department of Pharmacognosy, Lithuanian University of Health Sciences, Sukileliu av. 13, LT-50162 Kaunas, Lithuania; Mindaugas.Liaudanskas@lsmu.lt (M.L.); farmakog@lsmuni.lt (V.J.); 2Institute of Horticulture, Lithuanian Research Centre for Agriculture and Forestry, Kauno str. 30, LT-54333 Babtai, Kaunas District, Lithuania; Darius.Kviklys@lammc.lt (D.K.); Dalia.Gelvonauskiene@lammc.lt (D.G.); 3Norwegian Institute of Bioeconomy Research-NIBIO Ullensvang, Postboks str. 11, NO-1431 Ås Lofthus, Norway

**Keywords:** apple, phenolic compounds, genetic resources, HPLC-DAD

## Abstract

As the interest in heirloom cultivars of apple trees, their fruit, and processed products is growing worldwide, studies of the qualitative and quantitative composition of biological compounds are important for the evaluation of the quality and nutritional properties of the apples. Studies on the variations in the chemical composition of phenolic compounds characterized by a versatile biological effect are important when researching the genetic heritage of the heirloom cultivars in order to increase the cultivation of such cultivars in orchards. A variation in the qualitative and quantitative composition of phenolic compounds was found in apple samples of cultivars included in the Lithuanian collection of genetic resources. By the high-performance liquid chromatography (HPLC) method flavan-3-ols (procyanidin B1, procyanidin B2, procyanidin C2, (+)-catechin and (−)-epicatechin), flavonols (rutin, hyperoside, quercitrin, isoquercitrin, reynoutrin and avicularin), chlorogenic acids and phloridzin were identified and quantified in fruit samples of heirloom apple cultivars grown in Lithuania. The highest sum of the identified phenolic compounds (3.82 ± 0.53 mg/g) was found in apple fruit samples of the ‘Koštelė’ cultivar

## 1. Introduction

Apple trees are among the old cultivated fruit trees in the world [1]. Domestic apple trees (*Malus domestica* Borkh.) were starting to be grown 4000–10,000 years ago in the orchards of Central Asia [2]. Until the end of the 19th century, domestic apple trees were grown in the orchards of manors and monasteries [3,4]. Later on, local farmers started growing apple trees of the traditional cultivars in their orchards. During this period, a number of traditional apple cultivars of genetic resource heritage were bred in Lithuania, including ‘Lietuvos pepinas’, ‘Montvilinis’, ‘Popierinis’, ‘Rudeninis dryžuotasis’, ‘Žemaičių grietininis’, etc. [3]. It is expedient to preserve these apple cultivars, as their value has not been assessed yet.

The results of studies on the qualitative and quantitative composition of the fruit samples of apple cultivars grown in industrial orchards have been presented in scientific literature, but research data on the chemical composition of the fruit of heirloom apple cultivars grown in genetic heritage collections are fragmented [5]. The qualitative and quantitative composition of phenolic compounds in apples grown in Lithuanian industrial orchards has been investigated [5]. Secondary metabolites are phenolic compounds such as flavanols, flavonols, hydroxycinnamic acids, dihydrochalcones and anthocyanins were identified in fruit of the heirloom apple cultivars grown in Portuguese and Polish orchards [6,7]. Fruit samples of heirloom apple cultivars grown in Italian orchards and collections were found to have higher levels of phenolic compounds compared to those found in samples of apples of cultivars grown in industrial orchards [8,9].

Phenolic compounds have a wide range of biological effects: they act as strong antioxidants, scavenging free radicals [9,10], inhibit inflammatory processes [11,12] and the multiplication of bacteria [13] and they also have anti-cancer [14,15] and anti-aging effects [16,17,18]. The recommended daily diet should include apples, their processed products, and dietary supplements for the prevention of diabetes, asthma, cancer, and neurodegenerative and chronic cardiovascular diseases [11,15,19].

Industrial horticulture has been developing highly intensively during recent decades [1]. New apple cultivars are bred in industrial orchards, taking into account the needs of consumers in different regions of the world [1,20]. The introduction of new apple cultivars by fruit tree breeders reduced the demand for heirloom cultivars of apples grown in orchards [1,21]. Numerous cultivars of apple-specific trait donors have been used to create new apple cultivars grown in industrial orchards. Therefore, the characteristics of fruit trees of these cultivars (adaptability to biotic and abiotic factors, fruit taste, texture of the flesh, etc.) based on genetic factors are “harmonized”. Consumers are missing the traditional cultivars due to the variety of their fruit taste, aroma, consistency and suitability for unique national heritage products. Many countries, including Lithuania, have signed the Convention on Biological Diversity [22], the TREATY agreement [23] and the Nagoya Protocol [24] to assess the genetic uniqueness, distinctiveness and risk of extinction of heirloom varieties of garden plants. The signatories of the mentioned documents are committed to collecting, preserving and researching the diversity and economic, biological and medical value of the genetic resources of heirloom varieties of agricultural plants. 

Currently, consumers are looking for high-quality products with a known composition and health benefits [25]. Research into the qualitative and quantitative composition of biologically active compounds is important in assessing the quality and nutritional properties of apples. In Lithuania, apple cultivars belonging to the collection of genetic resources are grown in private orchards. In order to study the genetic heritage of heirloom cultivars with the aim of promoting their cultivation in orchards, it is important to carry out detailed studies on the variability of the chemical composition of phenolic compounds with biological effects. Data on qualitative and quantitative chemical composition will provide new scientific knowledge about variability of the qualitative and quantitative composition of biologically active compounds in the fruit of apple trees of heirloom cultivars.

The aim of the study was to investigate variability of the qualitative and quantitative composition of phenolic compounds in the samples of heirloom apple cultivars, to substantiate the cultivation of heirloom cultivars in Lithuanian orchards, and to preserve the genetic heritage of heirloom cultivars.

## 2. Results and Discussion

### 2.1. Qualitative and Quantitative Analysis of Phenolic Compounds of Apple of Heirloom Cultivars

Scientific literature provides data on the variability of the chemical composition of phenolic compounds in fruit samples of apple trees grown in industrial orchards [26]. The total amount of phenolic compounds found in fruit samples of apple cultivars ‘Aldas’, ‘Auksis’, ‘Ligol’, and ‘Šampion’ grown in Lithuanian industrial orchards ranged from 1.64 mg/g to 5.75 mg/g [27,28]. In Lithuania as well as in many other countries, collections and private orchards contain heirloom apple cultivars that form the heritage of genetic resources. Fruit of the heirloom apple cultivars are characterized by a high nutritional value and are of different shapes, colors, sizes and organoleptic properties (crispness, juiciness, and sweetness) [29]. Studies of the chemical composition of the fruit of heirloom apple cultivars are patchy.

The sum of the identified phenolic compounds in the samples of heirloom apple cultivars included in the collection of the Lithuanian heritage of genetic resources varied from 0.15 ± 0.01 mg/g to 3.82 ± 0.53 mg/g (Figure 1). The highest sum of the identified phenolic compounds (3.82 ± 0.53 mg/g) was detected in samples of the ‘Koštelė’ apple cultivar, which differed statistically significantly from the amount of these compounds detected in apple fruit samples of the other studied cultivars (*p* < 0.05) (Figure 1). The lowest content of the sum of the identified phenolic compounds (0.15 ± 0.01 mg/g) was found in fruit samples of the ‘Birutės pepinas’ cultivar, and it did not differ statistically significantly from that detected in fruit samples of ‘Beržininkų ananasinis’, ‘Danų karalienė Luiza’, ‘Montvilinis’, ‘Panemunės baltasis’, ‘Pilkasis alyvinis’, ‘Raudonasis alyvinis’ or ‘Žemaičių grietininis’ apple cultivars (Figure 1).

Studies of the fruit samples of heirloom apple cultivars grown in the orchards of the Marche region of Italy showed that the total amount of phenolic compounds ranged from 0.82 mg/g to 3.60 mg/g [30]. The total amount of phenolic compounds in fruit samples of heirloom apple cultivars grown in Brazilian orchards ranged from 0.46 mg/g to 1.58 mg/g [31]. Meanwhile, the total amount of phenolic compounds in fruit samples of heirloom apple cultivars grown in orchards of the Piedmont region of Italy ranged from 0.45 mg/g to 5.00 mg/g [8]. The sum of the identified phenolic compounds in fruit samples of heirloom apple cultivars grown in Lithuanian orchards was higher than that detected in fruit samples of the heirloom apple cultivars grown in the orchards of the Italian Marche region or Brazil, but lower than that found in fruit samples of the heirloom apple cultivars grown in the orchards of the Italian region of Piedmont.

#### 2.1.1. Variation of the Amount of Flavan-3-ols

Flavan-3-ols (procyanidin B1, procyanidin B2, procyanidin C2, (+)-catechin, and (−)-epicatechin) identified in fruit samples of heirloom apple cultivars grown in Lithuania accounted for 30% of the total amount of the identified and quantified phenolic compounds. The flavan-3-ol content in fruit samples of heirloom apple cultivars ranged from 0.03 ± 0.001 mg/g to 1.40 ± 0.05 mg/g (Figure 2). The flavan-3-ol content in fruit samples of heirloom apple cultivars grown in Croatian orchards was found to vary from 0.02 mg/g to 0.69 mg/g [26]. The flavan-3-ol content in fruit samples of heirloom apple cultivars grown in Italian orchards ranged from 0.02 mg/g to 0.66 mg/g [32]. The flavan-3-ol content in fruit samples of heirloom red-fleshed apple cultivars grown in Spanish orchards was found to range from 0.016 mg/g to 0.022 mg/g, while the flavan-3-ol content in samples of heirloom white-fleshed apples ranged from 0.09 mg/g to 0.20 mg/g [33]. The content of flavan-3-ols in fruit samples of heirloom apple cultivars of the Lithuanian genetic heritage collection was found to be higher than that in fruit samples of heirloom apple cultivars grown in Croatian, Italian, or Spanish orchards. 

The predominant compound of the flavan-3-ol group in fruit samples of heirloom apple cultivars grown in Lithuania was (−)-epicatechin. The highest amount of (−)-epicatechin (0.53 ± 0.01 mg/g) was found in fruit samples of the ‘Koštelė’ cultivar, and it was statistically significantly (*p* < 0.05) different from the amount of (−)-epicatechin detected in fruit samples of other apple cultivars (Figure 2). The content of (−)-epicatechin in fruit samples of heirloom apple cultivars grown in Polish orchards was found to range from 0.05 mg/g to 2.79 mg/g [34]. The amount of (−)-epicatechin in fruit samples of heirloom apple cultivars grown in Italian orchards ranged from 0.09 mg/g to 0.53 mg/g [8]. The data obtained by Polish and Italian researchers corroborate the results of our research.

The highest amounts of (+)-catechin (0.14 ± 0.01 mg/g) were found in fruit samples of the ‘Koštelė’ apple cultivar (Figure 2). The analysis of the fruit samples of heirloom apple cultivars showed a statistically significant (*p* < 0.05) difference in (+)-catechin content. The amount of (+)-catechin in fruit samples of heirloom apple cultivars grown in Italian orchards ranged from 0.02 mg/g to 0.05 mg/g [9]. Meanwhile, the amount of (+)-catechin in fruit samples of heirloom apple cultivars grown in Polish orchards ranged from 0.01 mg/g to 0.72 mg/g [35]. Fruit samples of heirloom apple cultivars grown in Lithuanian orchards were found to contain higher amounts of (+)-catechin, compared to that in fruit samples of heirloom apple cultivars grown in Italian orchards, but lower than that found in fruit samples of heirloom apple cultivars grown in Polish orchards. Procyanidins are among the most common flavan-3-ols found in samples of heirloom apple cultivars [8]. The highest amounts of procyanidin B2 (0.72 ± 0.18 mg/g), procyanidin C2 (0.14 ± 0.03 mg/g) and procyanidin B1 (0.06 ± 0.01 mg/g) were found in apple fruit samples of the ‘Virginijos rožinis’ cultivar (*p* < 0.05) (Figure 2). Procyanidin B2 predominated among the procyanidins identified and quantified in apple samples of heirloom cultivars grown in Lithuania. The amount of procyanidin B2 in fruit samples of heirloom apple cultivars grown in Polish orchards ranged from 0.07 mg/g to 2.00 mg/g [35]. Meanwhile, the amount of procyanidin B2 in fruit samples of heirloom apple cultivars grown in Italian orchards ranged from 0.018 mg/g to 2.09 mg/g [8]. The amount of procyanidin C1 in fruit samples of heirloom apple cultivars grown in Polish orchards ranged from 0.0006 mg/g to 0.97 mg/g [35]. The amount of procyanidin B1 in fruit samples of heirloom apple cultivars grown in Italian orchards ranged from 0.005 mg/g to 0.34 mg/g [8] and from 0.006 mg/g to 0.014 mg/g [32]. The results of studies on the variability of procyanidin content in fruit samples of heirloom apple cultivars from the collection of the Lithuanian heritage of genetic resources are confirmed by the data of research conducted by Polish and Italian scientists.

According to their amount in fruit samples of heirloom apple cultivars grown in Lithuanian orchards, the compounds of the flavan-3-ol group can be arranged in the following order: (−)-epicatechin>procyanidin B2>procyanidin C2>(+)-catechin>procyanidin B1. Studies of the qualitative and quantitative composition of compounds of the flavan-3-ol group are valuable due to the antioxidant properties of this group of compounds [10] and their glucose regulating effects [34].

#### 2.1.2. Variation of the Amount of Flavonols

The following flavonols were identified and quantified in fruit samples of heirloom apple cultivars from the collection of the Lithuanian heritage of genetic resources: rutin, hyperoside, quercitrin, isoquercitrin, reynoutrin, and avicularin. They comprised 13% of all the identified and quantified phenolic compounds. The content of flavonols in fruit samples of heirloom apple cultivars ranged from 0.04 ± 0.001 mg/g to 0.47 ± 0.12 mg/g (Figure 3). Studies of fruit samples of heirloom apple cultivars grown in Croatian orchards showed that flavonol levels ranged from 0.20 mg/g to 1.22 mg/g [26]. Meanwhile, flavonol levels in fruit samples of heirloom apple cultivars grown in Austrian orchards ranged from 0.67 mg/g to 5.66 mg/g [19]. The results of our study are corroborated by research data obtained by Croatian and Polish researchers.

Hyperoside was the predominant compound of the flavonols group in fruit samples of heirloom apple cultivars grown in Lithuania. The highest amount of hyperoside (0.19 ± 0.01 mg/g) was found in apple fruit samples of the ‘Koštelė’ cultivar, and it was statistically significantly different from hyperoside content in apple samples of other studied cultivars (*p* < 0.05) (Figure 3). The amount of hyperoside in fruit samples of heirloom apple cultivars grown in Italian orchards ranged from 0.0003 mg/g to 0.002 mg/g [8]. The amount of hyperoside found in fruit samples of heirloom apple cultivars grown in Lithuania was higher than that detected in fruit samples of heirloom apple cultivars grown in Italian orchards. The highest amount of avicularin (0.11 ± 0.01 mg/g) was found in apple fruit samples of the ‘Lietuvos pepinas’ cultivar (Figure 3). Avicularin content differed statistically significantly between fruit samples of heirloom apple cultivars (*p* < 0.05). The highest amount of quercitrin (0.06 ± 0.002 mg/g) was found in apple fruit samples of the ‘Koštelė’ cultivar, and it was statistically significantly different from that found in apple samples of other cultivars (*p* < 0.05) (Figure 3). The amount of quercitrin found in fruit samples of heirloom apple cultivars grown in Italian orchards ranged from 0.005 mg/g to 0.043 mg/g [8]. The amount of quercitrin in fruit samples of heirloom apple cultivars grown in Lithuanian orchards was higher, compared to the amount found in fruit samples of heirloom apple cultivars grown in Italian orchards. The highest amounts of isoquercitrin (0.06 ± 0.002 mg/g) and reynoutrin (0.04 ± 0.002 mg/g) were found in apple fruit samples of the ‘Lietuvos pepinas’ cultivar (Figure 3). The amount of isoquercitrin differed statistically significantly between fruit samples of differed heirloom apple cultivars (*p* < 0.05). There was no statistically significant difference in the amount of reynoutrin between apple fruit samples of the ‘Sierinka’ and ‘Koštelė’ cultivars (*p* > 0.05). Among the fruit samples of heirloom apple cultivars grown in Lithuania, the highest amount of rutin (0.04 ± 0.002 mg/g) was found in fruit samples of the ‘Paprastasis antaninis’ cultivar, and it did not differ statistically significantly from that found in apple samples of ‘Pilkasis alyvinis’, ‘Beržininkų ananasinis’, ‘Golden russet’, ‘Jono pepinas’, ‘Koštelė’, ‘Sierinka’, ‘Tabokinė’, ‘Baltasis alyvinis’, or ‘Lietuvos pepinas’ cultivars (*p* > 0.05) (Figure 3). The amount of rutin found in fruit samples of heirloom apple cultivars belonging to the collection of the Lithuanian heritage of genetic resources was higher compared to the amount of rutin (0.004 mg/g) found in fruit samples of heirloom apple cultivars grown in Italian orchards [9].

According to their amount in fruit samples of heirloom apple cultivars belonging to the collection of the Lithuanian heritage of genetic resources, the compounds of the flavonols group can be arranged in the following order: hyperoside>avicularin>quercitrin>isoquercitrin>reynoutrin>rutin. Studies of the qualitative and quantitative composition of the compounds of the flavonols group are important due to their antioxidant [34], anti-inflammatory [11], and antiallergic properties [36].

#### 2.1.3. Variation of the Amount of Chlorogenic Acid

Phenolcarboxylic acids are an important group of secondary metabolites in apples. It is important to determine the variability of their quantitative composition in fruit samples of heirloom apple cultivars. Chlorogenic acid was the predominant compound, making up 50% of the total amount of the identified and quantified phenolic compounds. The amount of chlorogenic acid in fruit samples of heirloom apple cultivars ranged from 0.01 ± 0.001 mg/g to 2.35 ± 0.03 mg/g (Figure 4).

The highest amount of chlorogenic acid (2.35 ± 0.03 mg/g) was detected in apple fruit samples of the ‘Lietuvos pepinas’ cultivar (Figure 4). The content of chlorogenic acid in fruit samples of heirloom apple cultivars differed statistically significantly (*p* < 0.05). The lowest amount of chlorogenic acid (0.01 ± 0.001 mg/g) was found in apple fruit samples of the ‘Montvilinis’ cultivar (Figure 4). The amount of chlorogenic acid in fruit samples of heirloom apple cultivars grown in orchards in the Tuscan region of Italy was found to vary from 0.12 mg/g to 0.63 mg/g [9]. Meanwhile, the amount of chlorogenic acid in fruit samples of heirloom apple cultivars grown in orchards in the Piedmont region of Italy ranged from 0.13 mg/g to 2.08 mg/g [8]. Fruit samples of heirloom apple cultivars from the collection of the Lithuanian heritage of genetic resources contained higher amounts of chlorogenic acid compared to those found in fruit samples of heirloom apple cultivars grown in Italian orchards.

#### 2.1.4. Variation of the Amount of Phloridzin

Compounds of the dihydrochalcone group are naturally prevalent in the vegetative and generative organs of plants of the apple (*Malus L*.) genus, while in other plant species, they are almost undetectable [33]. Fruit samples of heirloom apple cultivars were found to contain the dihydrochalcone group compound phloridzin, which accounted for 7% of the total amount of the identified and quantified phenolic compounds. The amount of phloridzin in apple samples ranged from 0.02 ± 0.002 mg/g to 0.30 ± 0.005 mg/g (Figure 5).

The highest content of phloridzin (0.30 ± 0.005 mg/g) was found in apple fruit samples of the ‘Golden russet’ cultivar (Figure 5), which did not differ statistically significantly only from the amounts found in apple fruit samples of the ‘Jono pepinas’ cultivar. The lowest amount of phloridzin (0.02 ± 0.002 mg/g) was found in apple fruit samples of the ‘Beržininkų ananasinis’ cultivar (Figure 5). The amount of phloridzin in fruit samples of heirloom apple cultivars grown in orchards in the Piedmont region of Italy was found to range from 0.001 mg/g to 0.26 mg/g [8]. The amount of phloridzin in fruit samples of heirloom apple cultivars grown in orchards in the Garfagnana region of Italy ranged from 0.01 mg/g to 0.05 mg/g [32]. Fruit samples of heirloom apple cultivars from the collection of the Lithuanian heritage of genetic resources contained higher amounts of phloridzin compared to those found in fruit samples of heirloom apple cultivars grown in Italian orchards.

### 2.2. Hierarchical Cluster Analysis of Phenolic Compounds of Apple of Heirloom Cultivars

Hierarchical cluster analysis of heirloom apple cultivars was performed, the results of which are presented in Figure 6. Based on the variability of the quantitative composition in apple samples of heirloom cultivars, phenolic compounds were distributed into clusters.

Fruit samples of heirloom apple cultivars assigned to cluster I (3, 4, 6, 7, 8, 14, 15, 17, 18, 21 and 22) were found to contain lower than average amounts of flavonols. Fruit samples of heirloom apple cultivars assigned to cluster II (1, 2, 5, 9, 12, 16 and 20) were found to contain average amounts of flavonols. Meanwhile, fruit samples of heirloom apple cultivars assigned to cluster III (10, 11, 13 and 19) were found to contain higher than average amounts of flavonols (Figure 6A). Fruit samples of heirloom apple cultivars assigned to cluster I (1, 2, 3, 4, 6, 7, 12, 14, 15, 16, 17, 20, 21 and 22) were found to contain lower than average amounts of chlorogenic acid. Fruit samples of the heirloom apple cultivar assigned to cluster II (18) had average amounts of chlorogenic acid. Meanwhile, fruit samples of heirloom apple cultivars assigned to cluster III (5, 8, 9 and 13) had higher than average amounts of chlorogenic acid. The highest amounts of chlorogenic acid were found in fruit samples of heirloom apple cultivars assigned to cluster IV (10, 11 and 19) (Figure 6B).

Fruit samples of heirloom apple cultivars assigned to cluster I (1, 2, 3, 4, 6, 11, 12, 14, 15, 16, 17, 18, 20 and 22) were found to contain lower than average amounts of (-)-epicatechin, (+)-catechin and phloridzin. Fruit samples of heirloom apple cultivars assigned to cluster II (8 and 9) were found to contain average amounts of (−)-epicatechin, (+)-catechin and phloridzin. Higher than average levels of (−−)-epicatechin, (+)-catechin and phloridzin were found in fruit samples of heirloom apple cultivars assigned to cluster III (5, 7, 13, 19 and 21). The highest levels of (−)-epicatechin, (+)-catechin and phloridzin were found in fruit samples of the heirloom apple cultivar assigned to cluster IV (10) (Figure 6C). Fruit samples of heirloom apple cultivars assigned to cluster I (1, 2, 5, 7, 14, 16, 18, 20 and 22) were found to contain average amounts of procyanidin B1, procyanidin B2 and procyanidin C2. Fruit samples of heirloom apple cultivars assigned to cluster II (3, 4, 6, 11, 12, 15 and 17) contained lower than average amounts of procyanidins. Higher than average amounts of procyanidins were found in fruit samples of heirloom apple cultivars assigned to cluster III (8, 9, 10, 13 and 19). The highest amounts of procyanidin B1, procyanidin B2, and procyanidin C2 were found in fruit samples of heirloom apple cultivars assigned to cluster IV (21) (Figure 6D).

### 2.3. Principal Component Analysis of Phenolic Compounds of Apple of Heirloom Cultivars

In this study, we analyzed the main components of phenolic compounds in fruit samples of heirloom apple cultivars. Two main components were used for the analysis, as they explain 80.19% of the total variability in the study data (Figure 7).

The amounts of isoquercitrin (0.939), hyperoside (0.930), avicularin (0.930), quercitrin (0.922), chlorogenic acid (0.902) and reynoutrin (0.815) strongly positively correlated with the first component, which describes 48.53% of the total data variability, while the correlation of the amounts of rutin (0.707) and phloridzin (0.691) with this component was strongly positive (Figure 7). The amounts of procyanidin C2 (0.889), procyanidin B1 (0.887), (−)-epicatechin (0.882), procyanidin B2 (0.878) and (+)-catechin (0.805) very strongly positively correlated with the second component, which describes 31.66% of the dispersion (Figure 7). 

Recently, there has been a growing interest in the genetic resources of heirloom cultivars, they are more widely grown, and there is an increasing number of studies on the qualitative and quantitative variability of the composition of their fruit. Fruit samples of some heirloom apple cultivars were found to have a richer quantitative and qualitative composition of phenolic compounds. Heirloom apple cultivars can be used for the selection of new apple cultivars. Apple trees of heirloom cultivars are becoming more popular, and higher levels of biologically active compounds are detected in their fruit [9]. Fruit samples of heirloom apple cultivars grown in Lithuania were found to contain 2.6 times higher amounts of flavonols, 7 times higher amounts of dihydrochalcones, 1.2 times lower amounts of phenolic acids and 1.5 times lower amounts of flavan-3-ols compared to those detected in fruit samples of heirloom apple cultivars grown in Polish orchards [34]. Quantitative differences can be explained by the competitive interaction between the enzymes anthocyanidin reductase and anthocyanidin synthase during flavonoid synthesis, which results in a slower synthesis of flavan-3-ols and their lower accumulation in apples [2].

The results of our study provided new knowledge about apple cultivars from the collection of the Lithuanian heritage of genetic resources and the variability of the qualitative and quantitative composition of the phenolic compounds found in their fruit. The highest sum of the identified phenolic compounds (3.82 ± 0.53 mg/g) was found in apple fruit samples of the ‘Koštelė’ cultivar. The sum of the identified phenolic compounds was higher than that (0.86 mg/g) found in fruit samples of heirloom apple cultivars grown in Germany [36]. Fruit samples of heirloom apple cultivars included in the collection of the Lithuanian heritage of genetic resources contained higher amounts of phenolic compounds compared to those detected in fruit samples of the ‘Jonagold’ cultivar grown in the orchards of Lhasa (Italy), Rokietnica (Poland), and Randwijk (the Netherlands) regions (respectively, 2.21 mg/g, 2.69 mg/g, and 3.81 mg/g). However, this amount was lower than that (4.76 mg/g) found in the samples of apples grown in the orchards of the Wieluń region of Poland [37]. In fruit samples of heirloom apple cultivars grown in Lithuania, chlorogenic acid comprised the greatest part of the phenolic compounds. The highest content of chlorogenic acid (2.35 ± 0.03 mg/g) was found in apple fruit samples of the ‘Lietuvos pepinas’ cultivar. Polish researchers indicated that chlorogenic acid might account for 64–94% of all the identified and quantified phenolic acids in apple fruit samples [34]. Chlorogenic acid in fruit and vegetables determines their sensory properties and has anti-mutagenic and antioxidant effects [34]. The compound phloridzin belonging to the dihydrochalcone group was found in fruit samples of heirloom apple cultivars. Its highest amounts (0.30 ± 0.005 mg/g) were found in apple fruit samples of the ‘Golden russet’ cultivar. Phloridzin is a biologically active compound with a wide range of biological effects. It regulates blood glucose levels [38,39], antioxidant and anti-aging effects [11,40]. Qualitative and quantitative analysis of dihydrochalcone group compounds is important, as they can be used as chemotaxonomic markers in the taxonomy of apple species as well as for the identification and quality assessment of apple products [6].

## 3. Materials and Methods

### 3.1. Plant Materials

The study included 22 heirloom apple cultivars, of which 21 (except for ‘Golden russet’) are included in the List of the National Plant Genetic Resources (Table 1).

The apple trees were grown in the Collection of the Apple Tree Genetic Resources at the Institute of Horticulture (in Babtai town), a division of the Lithuanian Research Centre for Agriculture and Forestry (henceforth, LAMMC). Coordinates: 55°60′ N, 23°48′ E. The study was conducted during 2019–2020.

### 3.2. Chemicals and Solvents

All solvents, reagents, and standards used were of analytical grade. Acetonitrile and acetic acid were obtained from Sigma-Aldrich GmbH (Buchs, Switzerland), ethanol was obtained from AB Stumbras (Kaunas, Lithuania), hyperoside, rutin, quercitrin, phloridzin, procyanidin B1, procyanidin B2 and chlorogenic acid standards were purchased from Extrasynthese (Genay, France), reynoutrin, (+)-catechin and (−)-epicatechin–from Sigma-Aldrich GmbH (Steinheim, Germany), and avicularin, procyanidin C1 and isoquercitrin–from Chromadex (Santa Ana, CA, USA). Purified deionized water used in the tests was prepared with the Milli-Q^®^ (Millipore, Bedford, MA, USA) water purification system.

### 3.3. Preparation of Samples

For the analysis, twenty apples at the optimal maturity stage were picked from different parts of the tree crown. Whole apples were immediately frozen in a freezer (at −35 °C) with air circulation. Subsequently, these frozen samples were lyophilized with a ZIRBUS sublimator 3 × 4 × 5/20 (ZIRBUS technology, Bad Grund, Germany) at a pressure of 0.01 mbar (condenser temperature: −85 °C). The lyophilized samples were ground to fine powder using a Retsch 200 mill electric grinder (Haan, Germany). Loss on drying before the analysis was determined by drying the apple lyophilisate in a laboratory drying oven to complete the evaporation of water and volatile compounds (temperature: 105 °C; the difference in weight between measurements: up to 0.01 g) and by calculating the difference in raw material weight before and after the drying. The data were recalculated for the absolute dry lyophilisate weight. The prepared apple samples were stored in dark, tightly closed glass vessels.

### 3.4. Preparation of the Phenolic Compounds

During the analysis of phenolic compounds, 2.5 g of lyophilizate powder (exact weight) was weighed, added to 30 mL of 70% (*v*/*v*) ethanol, and extracted in a Sonorex Digital 10 P ultrasonic bath (Bandelin Electronic GmbH & Co. KG, Berlin, Germany) at room temperature for 20 min. The obtained extract was filtered through a paper filter, and the residue on the filter was washed with 70% (*v*/*v*) ethanol in a 50-mL flask until the exact volume was reached. The conditions of the extraction were chosen based on the results of the tests for setting the extraction conditions.

### 3.5. Qualitative and Quantitative Analysis by HPLC–PDA Method

The qualitative and quantitative HPLC analysis of phenolic compounds was performed with a Waters 2998 PDA detector (Waters, Milford, CT, USA). Chromatographic separations were carried out by using a YMC-Pack ODS-A (5 μm, C18, 250 × 4.6 mm i.d.) column. The column was operated at a constant temperature of 25 °C. The volume of the analyzed extract was 10 μL. The flow rate was 1 mL/min. The mobile phase consisted of 2% (*v*/*v*) acetic acid (solvent A) and acetonitrile (solvent B). Gradient variation: 0–30 min 3–15% B, 30–45 min 15–25% B, 45–50 min 25–50% B, and 50–55 min 50–95% B. For the quantitative analysis, the calibration curves were obtained by injecting the known concentrations of different standard compounds. All the identified phenolic compounds were quantified at λ = 200–400 nm wavelength [5].

### 3.6. Statistical Analysis

The statistical analysis of the study data was performed by using Microsoft Office Excel 2013 (Microsoft, Redmond, WA, USA) and SPSS 25.0 (SPSS Inc., Chicago, IL, USA) computer software. All the results obtained during the ESC analysis were presented as means of three consecutive test results and standard deviations. To evaluate the variance in the quantitative composition, we calculated the coefficient of variation. Univariate analysis of variance (ANOVA) was applied in order to determine whether the differences between the compared data were statistically significant. The hypothesis about the equality of variances was verified by applying Levine’s test. If the variances of independent variables were found to be equal, Tukey’s multiple comparison test was used. The differences were regarded as statistically significant at *p* < 0.05. The comparison of the chemical composition between the apple fruit samples of the studied heirloom cultivars was carried out by applying the hierarchical cluster analysis, using the squared Euclidean distance. Principal component analysis was performed as well.

## 4. Conclusions

Apple trees of heirloom cultivars are valuable from the genetic aspect in the selection of new fruit tree cultivars. Their fruit are a source of biologically active compounds and can be used in the development and production of new innovative dietary supplements and medicinal cosmetic products. Apple trees of the heirloom cultivars ‘Koštelė’, ‘Lietuvos pepinas’, ‘Paprastasis antaninis’, ‘Virginijos rožinis’ and ‘Sierinka’ grown in the Collection of the Apple Tree Genetic Resources at the Institute of Horticulture of the Lithuanian Research Center for Agriculture and Forestry are not suitable for growing in industrial gardens due to their low fruit yield, poor external quality, small size, and susceptibility to disease. In amateur gardens, growing apples of heirloom cultivars is promising due to their higher content of bioactive substances. Our phytochemical studies of heirloom apple cultivars provide valuable scientific knowledge on the variability of the qualitative and quantitative composition of phenolic compounds. The results of our study will enable a wider cultivation of heirloom apple cultivars in gardens and collections and will help consumers to obtain and use apples with a known chemical composition of phenolic compounds, which determine the use of apples in the healthy food chain and the development of innovative food products.

## Figures and Tables

**Figure 1 molecules-25-05263-f001:**
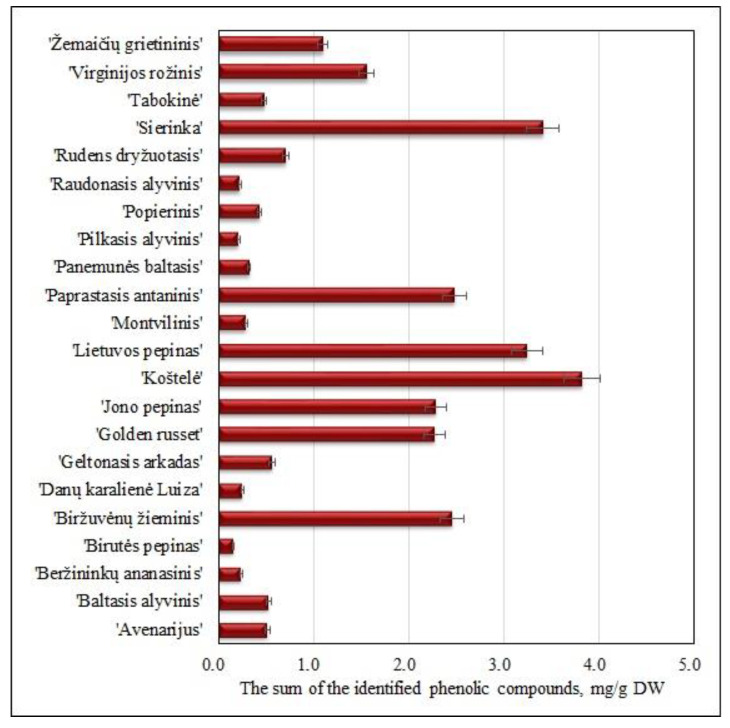
Variability of the sum of the identified phenolic compounds in fruit samples of heirloom apple cultivars.

**Figure 2 molecules-25-05263-f002:**
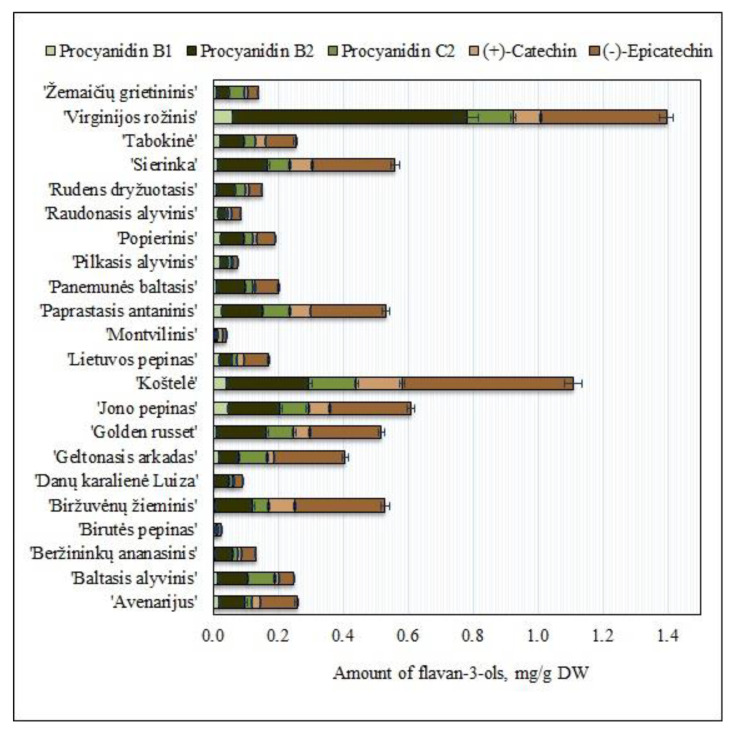
Variability of the amount of flavan-3-ols in fruit samples of heirloom apple cultivars.

**Figure 3 molecules-25-05263-f003:**
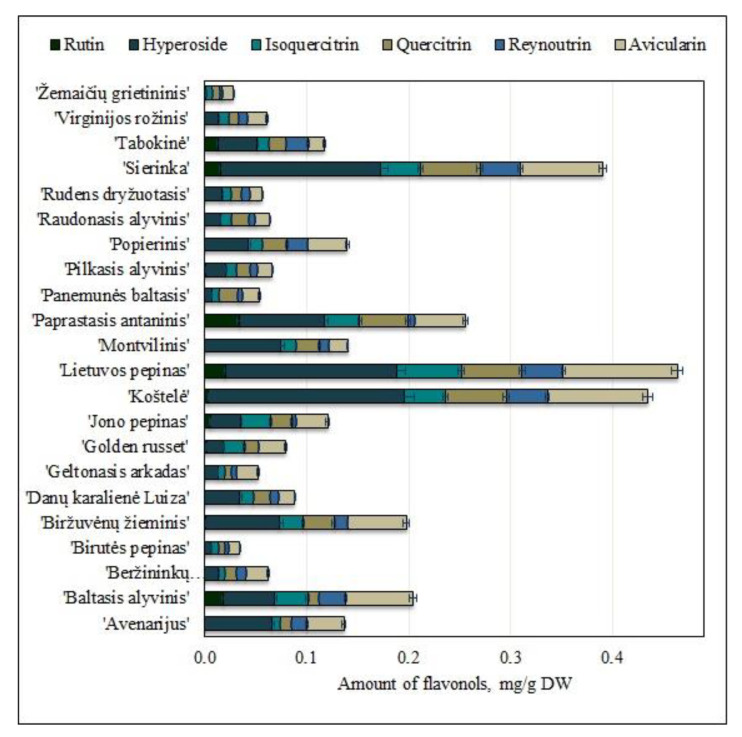
Variability of the amount of flavonols in fruit samples of heirloom apple cultivars.

**Figure 4 molecules-25-05263-f004:**
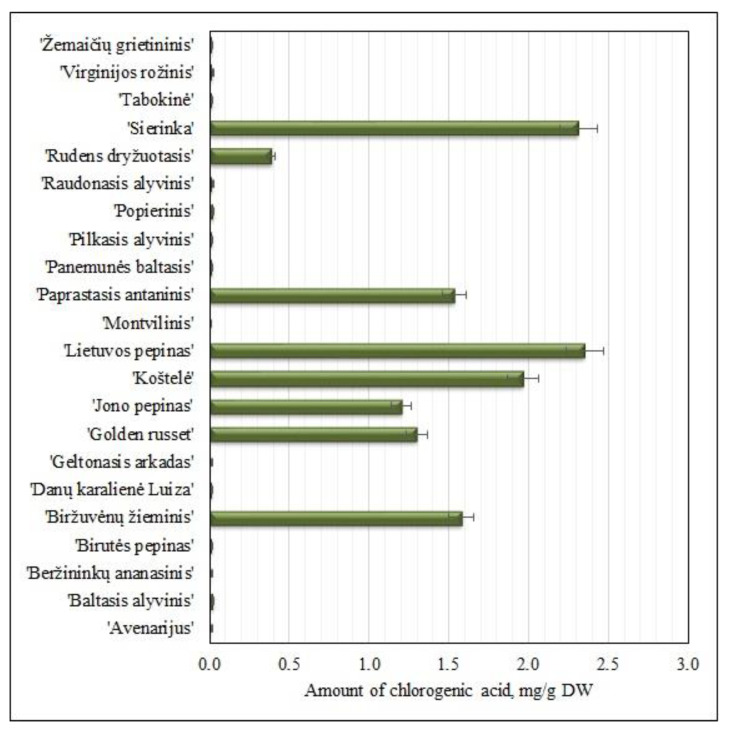
Variability of the amount of chlorogenic acid in fruit samples of heirloom apple cultivars.

**Figure 5 molecules-25-05263-f005:**
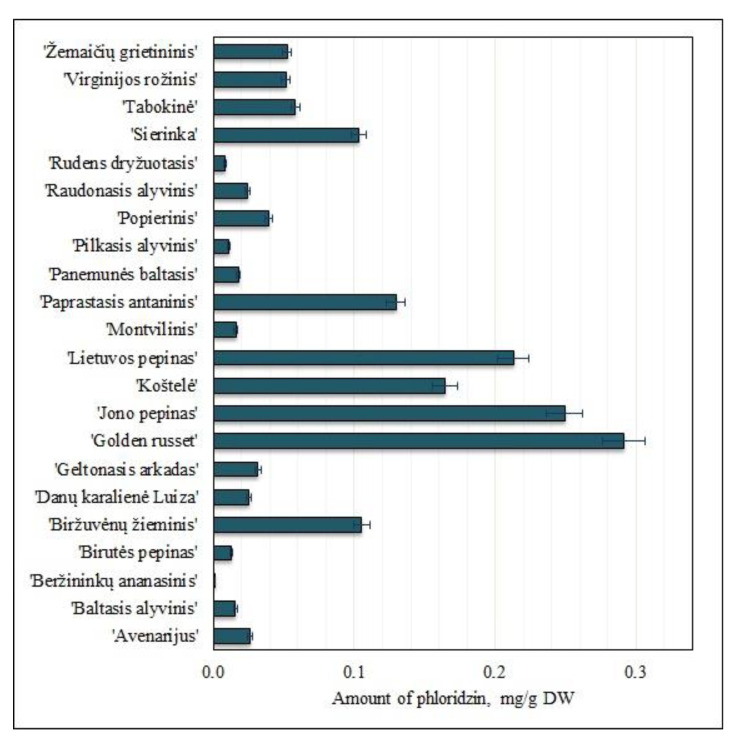
Variability of the amount of phloridzin in fruit samples of heirloom apple cultivars.

**Figure 6 molecules-25-05263-f006:**
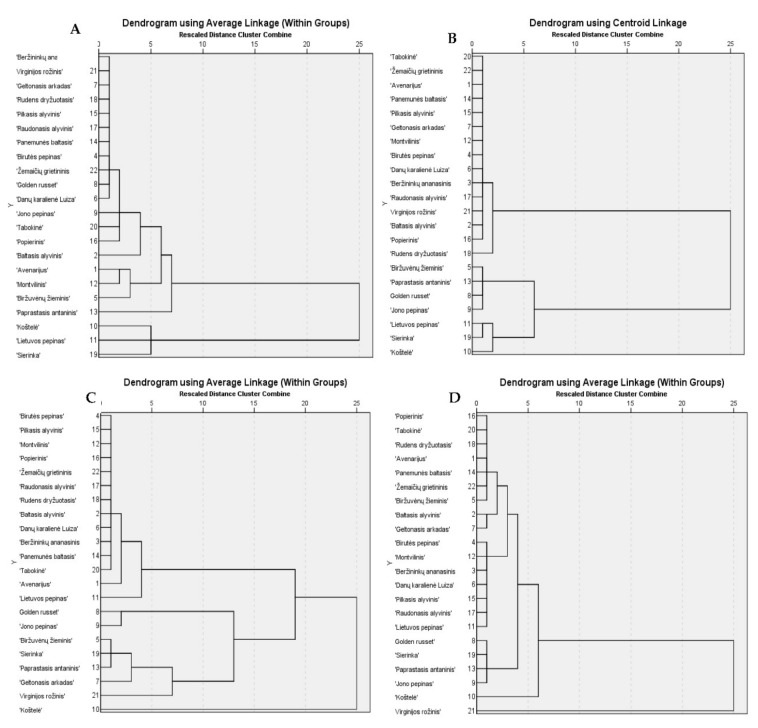
Dendrogram of the similarity of apple samples in terms of the amounts of phenolic compounds. Flavonols were distributed into three clusters (**A**), chlorogenic acid was distributed into four clusters (**B**), (−)-epicatechin, (+)-catechin, and phloridzin were distributed into four clusters (**C**), and compounds of the procyanidin group were distributed into four clusters (**D**).

**Figure 7 molecules-25-05263-f007:**
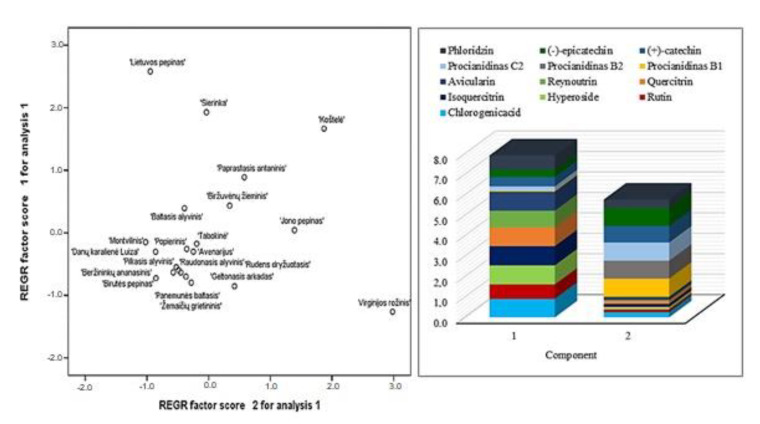
Analysis of the main components of phenolic compounds in apple samples.

**Table 1 molecules-25-05263-t001:** Origin and properties of the heirloom apple cultivars of Lithuania [4,41].

No.	Apple Cultivar	Year of Release, Finding, or Description, and Country	Other Exclusive Characteristics
1.	‘Avenarijus’	1886, Russia, SC	Skin greenish-yellow, flesh pink, sweet; susceptible to canker
2.	‘Baltasis alyvinis’	1848, Russia, SC	Skin yellow, flesh white, aromatic; susceptible to scab
3.	‘Beržininkų ananasinis’	1886, Lithuania, AC	Skin yellow, flesh crispy, aromatic; scab-resistant
4.	‘Birutės pepinas’	1941, Lithuania, AC	Skin reddish- white, flesh white, with suspicion of wine; susceptible to scab
5.	‘Biržuvėnų žieminis’	Lithuania, WC	Skin yellow, sweet; scab-resistant
6.	‘Danų karalienė Luiza’	1878, Denmark, WC	Skin covered with rust grid, flesh creamy yellow; scab-resistant
7.	‘Geltonasis arkadas’	XIX, Russia, SC	Skin yellow, sweet, sometimes astringent; susceptible to scab.
8.	‘Golden russet’	1800-1849, USA, WC	Skin strong russet, flesh creamy yellow; resistant to scab
9.	‘Jono pepinas’	XIX, Lithuania, WC	Flesh firm, yellow; scab-resistant
10.	‘Koštelė’	XIX, Poland, WC	Skin yellow, sweet, flesh firm, creamy; scab-resistant
11.	‘Lietuvos pepinas’	XVIII, Lithuania, WC	Skin yellow, vinous taste, flesh white; susceptible to scab
12.	‘Montvilinis’	1879, Lithuania, WC	Skin yellow, aromatic; scab-resistant
13.	‘Paprastasis antaninis’	XVIII, Russia, AC	Skin greenish-yellow, acidic, very aromatic; moderately scab-resistant
14.	‘Panemunės baltasis’	1939, Lithuania, AC	Skin greenish-yellow, flesh white, waxed; scab-resistant
15.	‘Pilkasis alyvinis’	1653, Russia, SC	Skin white-yellow, flesh white; susceptible to scab
16.	‘Popierinis’	1852, Lithuania or Latvia, SC	Skin white-yellow, flesh white; susceptible to scab
17.	‘Raudonasis alyvinis’	XVIII, Russia, SC	Skin reddish- white, aromatic, susceptible to scab
18.	‘Rudens dryžuotasis’	1870, Baltic countries, AC	Skin reddish-white, vinous taste, flesh pinkish; moderately scab-resistant
19.	‘Sierinka’	1860, Baltic countries, AC	Skin greenish-yellow, fragrant with characteristic aroma, susceptible to canker; moderately scab-resistant
20.	‘Tabokinė’	XIX, Baltic countries, WC	Skin reddish-yellow, bitter-sweet, bitterness weakens by spring; scab-resistant
21.	‘Virginijos rožinis’	1816, Europe, SC	Skin reddish-white, vinous taste; susceptible to scab
22.	‘Žemaičių grietininis’	XIX, Lithuania, SC	Skin white-yellow, flesh white; moderately scab-resistant

SC–summer cultivar, AC–autumn cultivar, WC–winter cultivar.

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
