# Peer review of "The Qualitative and Quantitative Compositions of Phenolic Compounds in Fruits of Lithuanian Heirloom Apple Cultivars"

_molecules, 2020, doi:10.3390/molecules25225263_

Round 1

Reviewer 1 Report

In my opinion the paper is not suitable for publication in its present form.

References are listed alphabetically, not according to the first appearance in the text.

The Authors did not describe how the total amount of phenolic compounds was assessed. Was it simply a sum of the quantified compounds? The explanation is necessary. Could the Authors exclude the presence of other phenolics (different from the purchased standards)? If not, the part of the text dealing with „total amount of phenolic compounds” should be withdrawn.

Lines 118 and 170 – „iki”?

Sentences from the lines 129-130 and 132-133 repeat the same information.

Was the chlorogenic acid the only phenolocarboxylic acid present in the analyzed material?

Figure 6 needs rescaling. The captions are illegible.

Line 311 – „pat”?

Section „Materials and Methods”

Oszmiański et al. [16] used an antioxidant (ascorbic acid) at the extraction step. Might it cause differences in quantities of the estimated compounds? Are the presented results fully suitable to compare with those obtained by the other teams?

Line 372 – „the calibration curve” – one curve?

Statements concerning biological activity of the studied compounds should be referred to the original experimental papers describing such activities.

Conclusions

Lines 395-405 are repetitions of the results.

The English language if understandable and basically correct, but the frequent repetition of the same phrases and words makes the reading of the text tedious.  

Author Response

In my opinion the paper is not suitable for publication in its present form.
1. References are listed alphabetically, not according to the first appearance in the text.
Response:
Thank you for the remark. The literature sources in the reference list have been rearranged according to the order of their appearance in the text.

2. The Authors did not describe how the total amount of phenolic compounds was assessed. Was it simply a sum of the quantified compounds? The explanation is necessary. Could the Authors exclude the presence of other phenolics (different from the purchased standards)? If not, the part of the text dealing with „total amount of phenolic compounds” should be withdrawn.
Response:
In this paper, the total amount of phenolic compounds in apples was the sum of the identified and quantified individual phenolic compounds. We agree with the reviewer’s remark that the apple extract contains more phenolic compounds, which we could not identify. In the article, we limited the analysis to the identified phenolic compounds and presented their arithmetic sum.

3. Lines 118 and 170 – „iki”?
Response:
Thank you for the remark. In lines 118 and 170, the word “iki” has been replaced with “to”.

4. Sentences from the lines 129-130 and 132-133 repeat the same information.
Response:
Thank you for your attentiveness. The sentence in lines 132-133 has been deleted to avoid repetition.

5. Was the chlorogenic acid the only phenolocarboxylic acid present in the analyzed material?
Response:
Only the chlorogenic acid was identified in the analyzed apple samples.

6. Figure 6 needs rescaling. The captions are illegible.
Response:
Thank you for the remark. Figure 6 has been replaced and rescaled. 

7. Line 311 – „pat”?
Response:
Lin 311, the word “pat” has been replaced with “part”.

8. Section „Materials and Methods”
Oszmiański et al. [16] used an antioxidant (ascorbic acid) at the extraction step. Might it cause differences in quantities of the estimated compounds? Are the presented results fully suitable to compare with those obtained by the other teams?
Response:
The antioxidant used during the extraction may have resulted in higher levels of compounds in the extracts. However, according to research data, the amount of individual biologically active compounds depends on a wide range of various environmental factors (climatic conditions, geographical locations, soil, care technologies of trees, time of picking, storage conditions, genetics, rootstock, and the cultivars of the apples). We believe that in this case it is expedient to compare the amounts of the compounds without singling out the extractant as the only factor determining the higher content of compounds in the extracts.

9. Line 372 – „the calibration curve” – one curve?
Response:
The sentence is incorrect. A separate calibration curve is required to determine the amount of each individual phenolic compound. In our studies, using the high-performance liquid chromatography technique, the calibration curves of the individual phenolic compounds were as follows: chlorogenic acid (Y=26800X-30200); rutin (Y=14700X-7660); hyperoside (Y=22300X-10100); isoquercitrin (Y=26600x-21000); quercitrin (Y=17700X5570); reynoutrin (Y=17700X+11500); avicularin (Y=23200X-11200); procyanidin B1(Y=7330X+5490); procyanidin B2 (Y=5580X+7150); procyanidin C2 (Y=5860X+359); (+)-catechin (Y=7000X-1870); (-)- epicatechin (Y=7530+119); and phloridzin (Y=18500X-7090). 

10. Statements concerning biological activity of the studied compounds should be referred to the original experimental papers describing such activities.
Response:
Thank you for the remark. However, we could not understand what the original experimental papers are. We would be grateful if you would elaborate on what we should improve specifically in this case.

11. Conclusions
Lines 395-405 are repetitions of the results.
Response:
Thank you for your attentiveness. We removed the sentences in lines 395-405

Reviewer 2 Report

1. This study investigates the compositions of phenolic compounds in some fruits from apple cultivars in Lithuanian. The data might be useful for apple breeders and the related industries.
2. What do you mean by "old" cultivars? Do you mean "classical" or "traditional" or "conventional" cultivars?
3. The descriptions for the fruit materials should be more detailed, such as the size, age, and physiological conditions of the trees as well as fruits. What do you mean by the optimal maturity stage? It should be defined well.
4. In the conclusion section, the authors should give a paragraph for perspectives. Maybe move one or two refined sentences from the last paragraph of the discussion section.
5. The order of the references in the text seems not satisfying, it should be used from 1 to 30, but not from 8, 5, 14, 16......
5. A suggestion for the title: Investigations Of The Qualitative And Quantitative Composition Of Phenolic Compounds In Fruit Of Lithuanian Old Apple Cultivars - The Qualitative and Quantitative Compositions of Phenolic Compounds in Fruits of Lithuanian Traditional Apple Cultivars
6. The main results such as the total amount of phenolic compounds should be compared with some popular international apple cultivars from literature in the discussion section to make the study not so local.

Author Response

This study investigates the compositions of phenolic compounds in some fruits from apple cultivars in Lithuanian. The data might be useful for apple breeders and the related industries.
1. What do you mean by "old" cultivars? Do you mean "classical" or "traditional" or "conventional" cultivars?
Response:
We agree that “old cultivar” is not exact description of historic or heirloom cultivars. Therefore we will use “heirloom cultivar” instead of “old cultivar”.

2. The descriptions for the fruit materials should be more detailed, such as the size, age, and physiological conditions of the trees as well as fruits. What do you mean by the optimal maturity stage? It should be defined well.
Response:
The orchard was planted in 2001, at 4 m × 3 m intervals. Tested cultivars were propagated on M.26 rootstock and were trained as slender spindles. Pest and disease management was carried out according to the rules of integrated plant protection. The study was carried out when orchard was in full production. Fruits were collected only from healthy and will developed trees. Fruits of each cultivar were harvested at optimal maturity stage that was defined by the development of typical to certain cultivar fruit size, ground colour, blush and starch test. 

3. In the conclusion section, the authors should give a paragraph for perspectives. Maybe move one or two refined sentences from the last paragraph of the discussion section.
Response:
We agree with your remark, and thus we have moved the last two sentences from the discussion part to the conclusions. 

4. The order of the references in the text seems not satisfying, it should be used from 1 to 30, but not from 8, 5, 14, 16..
Response:
Thank you for the remark. The literature sources in the reference list have been rearranged according to the order of their appearance in the text. 

5. A suggestion for the title: Investigations Of The Qualitative And Quantitative Composition Of Phenolic Compounds In Fruit Of Lithuanian Old Apple Cultivars - The Qualitative and Quantitative Compositions of Phenolic Compounds in Fruits of Lithuanian Traditional Apple Cultivars
Response:
We thank you and agree with your remark, changing the title of the article into The Qualitative and Quantitative Compositions of Phenolic Compounds in Fruits of Lithuanian Heirloom Apple Cultivar.

6. The main results such as the total amount of phenolic compounds should be compared with some popular international apple cultivars from literature in the discussion section to make the study not so local.
Response:
We have added the following new information to the discussion section: “The total amount of phenolic compounds was higher than that (0.86 mg/g) found in fruit samples of traditional apple cultivars grown in Germany [30]. Fruit samples of traditional apple cultivars included in the collection of the Lithuanian heritage
of genetic resources contained higher amounts of phenolic compounds compared to those detected in fruit samples of the 'Jonagold' cultivar grown in the orchards of Lhasa (Italy), Rokietnica (Poland), and Randwijk (the Netherlands) regions (respectively, 2.21 mg/g, 2.69 mg/g, and 3.81 mg/g). However, this amount was lower than that (4.76 mg/g) found in the samples of apples grown in the orchards of the Wieluń region of Poland [31].”

Reviewer 3 Report

the manuscript is clear and written without errors, however the content in terms of bringing interesting and relevant results to science seems to me a little poor.
The images in figure 6, the dendrograms cannot be read.
The cultivars used in the study are compared with cultivars from different areas of Italy and Brazil, as for phenols (only), but there is no information on the genetic data of distance or proximity of these different cultivars, there is no edaphoclimatic information of the areas where cultivars were studied. For the information to have relevance there were several other variables to take into account in the study. The goal is poor in terms of publication.

Author Response

The manuscript is clear and written without errors, however the content in terms of bringing interesting and relevant results to science seems to me a little poor.
1. The images in figure 6, the dendrograms cannot be read.
Response:
Thank you for the remark. Figure 6 has been replaced and rescaled.

2. The cultivars used in the study are compared with cultivars from different areas of Italy and Brazil, as for phenols (only), but there is no information on the genetic data of distance or proximity of these different cultivars, there is no edaphoclimatic information of the areas where cultivars were studied. For the information to have relevance there were several other variables to take into account in the study. The goal is poor in terms of publication.
Response:
Heirloom apple cultivars are different in different countries and regions due to different ways of their introduction, adaptation to local climate conditions and prevailing growing traditions. We studied apple cultivars that were cultivated in Lithuania from 19th century or introduced from neighbouring countries. Due
to different genetic background of heirloom cultivars in various countries we can compare only data of the whole collections gathered in different countries. We agree that different factors (climate, soil, management etc.) can affect the accumulation of bioactive compounds. But it is impossible to define effect of climate conditions when not the same cultivars were studied in different countries.

Round 2

Reviewer 1 Report

Pages 2-3 and further - "The total amount of phenolic compounds" should be replaced by "The sum of the identified phenolic compounds".

As to the citations of the original experimental works;

Example - iines 49-54:

Phenolic compounds ...inhibit inflammatory processes [6]. The reference is a review on potential use of bioactive compounds from apples in dermal formulations. It is not an experimental work on anti-inflammatory activity of phenolics and even not the review on an anti-inflammatory activity of phenolics. You should cite Pastore et al. 2009 (the paper cited by 6).

...multiplication of bacteria and viruses [10]. Bai et al. 2013 did their job properly, and after the statement concerning antibacterial and anticarcinogenic activity of the apple polyphenols cited two original experimental works: Pastene et al., 2009 and McCann et al., 2007.

You should give the credit to the people who experimentally proved activity of the compounds.

...strong anti-cancer and anti-aging effects [11]. Is it a scientifically proven fact or just gossip? 11 deals with a quite different matter. Moreover, the Authors of 11 referred only to the reviews or introductions of the other papers not strictly related with the investigation on anti-cancer and anti-aging activities. Please, give us a proof of anti-aging activity.

Sometimes the less is better. Rely only on the experimentally proven data. The same is valid for every statement concerning biological activity of the natural compounds.

Author Response

Thank you for the remark.

In the article, we replaced "the total amount of phenolic compounds" to "the sum of the identified phenolic compounds".

ROS are byproducts of metabolism, physiologically and continuously generated in the mithochondria, and also one of the main targets in harmful effects. SOD is anti-oxidative stress gene that takes part in free-radical scavening. Oxidative alterations to biomolecules increase with age, and are an obvious outcome of redox imbalance. The phenolic compounds act as natural antioxidants and bind free radicals, inhibit the products of their reactions, and stimulate the synthesis of antioxidant enzymes such as SOD, which prevent oxidative stress-induced lipid damage to molecular structures of the body such as DNA. It is hence expected that decreasing ROS production or increasing antioxidant defenses has anti-aging effects.

The citation about biological effect of phenolic compounds in lines 49-53 has been corrected with new references: …. Phenolic compounds have a wide range of biological effects: they act as strong antioxidants, scavenging free radicals [9-10], inhibit inflammatory processes [11-12], and the multiplication of bacteria [13], and they also have anti-cancer [14-15] and anti-aging effects [16-18]. The recommended daily diet should include apples, their processed products, and dietary supplements for the prevention of diabetes, asthma, cancer, and neurodegenerative and chronic cardiovascular diseases [11, 15, 19].

In lines 163-164 has been replaced ….. the antioxidant properties of this group of compounds [10] and their glucose, glycogen and lipid regulating effects [34].

In lines 207-208 has been improved….antioxidant properties [34], anti-inflammatory [11], and antiallergic [36].

In lines 326-327 has been replaced ….. It regulates blood glucose levels [38-39], antioxidant and anti-aging effect [11, 40].

We have added new references [10-18] and [38-40].

Reviewer 3 Report

although the manuscript has been improved it is still not very innovative and shallow

Author Response

Thank you for your remark and opinion.